# Culex Flavivirus Isolation from Naturally Infected Mosquitoes Trapped at Rio de Janeiro City, Brazil

**DOI:** 10.3390/insects13050477

**Published:** 2022-05-19

**Authors:** Cinthya Amaral, Daniel Câmara, Tiago Salles, Marcelo Damião Meneses, Carlla de Araújo-Silva, Vanessa Dias, Fábio da Costa, Lúcio Caldas, Renata Azevedo

**Affiliations:** 1Departamento de Virologia, Instituto de Microbiologia Paulo de Góes, Universidade Federal do Rio de Janeiro, Av. Carlos Chagas Filho, 373, Rio de Janeiro CEP 21941-970, RJ, Brazil; cinthya.domingues@gmail.com (C.A.); tiagosouzasalles@gmail.com (T.S.); marcelomeneses@micro.ufrj.br (M.D.M.); vanessazaquieu@gmail.com (V.D.); fabio_burack@hotmail.com (F.d.C.); 2Laboratório de Mosquitos Transmissores de Hematozoários-LATHEMA, Instituto Oswaldo Cruz, Fundação Oswaldo Cruz, Av. Brasil, 4365, Rio de Janeiro CEP 21040-360, RJ, Brazil; dcpchamber@gmail.com; 3Núcleo Operacional Sentinela de Mosquitos Vetores-Nosmove, Fundação Oswaldo Cruz Av. Brasil, 4365, Rio de Janeiro CEP 21040-360, RJ, Brazil; 4Laboratório de Ultraestrutura Celular Hertha Meyer, Universidade Federal do Rio de Janeiro, Av. Carlos Chagas Filho, 373, Rio de Janeiro CEP 21941-590, RJ, Brazil; carllaaraujo@biof.ufrj.br (C.d.A.-S.); lucio@biof.ufrj.br (L.C.); 5Intituto Nacional de Ciência e Tecnologia para Biologia Estrutural e Bioimagem, Av. Carlos Chagas Filho, 373, Rio de Janeiro CEP 21941-590, RJ, Brazil

**Keywords:** *Culex Flavivirus*, insect-specific virus, virus isolation, flavivirus, *Culex* species

## Abstract

**Simple Summary:**

The Flavivirus genus groups a wide range of species capable of infecting vertebrates and invertebrates, both terrestrial and aquatic. According to phylogenetic analyses, the flavivirus genomes cluster into three main branches; the first one containing viruses that infect vertebrates, also called arboviruses; the second called arbovirus-affiliated insect-specific flaviviruses or dual-host insect-specific flaviviruses (dISF), that preserve genomic similarity with arboviruses, but its replication is apparently restricted to invertebrates and insect-specific classical flaviviruses (ISF), with infection restricted to invertebrates. *Culex Flavivirus* (CxFV) *is* a classical insect-specific virus, which has aroused interest after the first indication that it can produce in nature superinfection exclusion of viruses of medical interest such as West Nile. Despite the detection of CxFV in different regions, CxFV ecology and the influence of co-circulation of arboviruses remains poorly understood. Therefore, our primary goals are to observe the occurrence of CxFV infection in mosquitoes trapped in an urban area of Rio de Janeiro, Brazil, characterize the virus circulation, and provide isolates. A prospective study was carried out for eight months on the Federal University of Rio de Janeiro campus trapping adult mosquitoes. The CxFV minimum infection rates were determined in this period, and the virus isolation process is fully described. Samples from this study were grouped into genotype 2, along with CxFV sequences from Latin America and Africa.

**Abstract:**

*Culex Flavivirus* (CxFV) is a classical insect-specific virus, which has aroused interest after the first indication that it can produce in nature superinfection exclusion of viruses of medical interest such as West Nile. Despite the detection of CxFV in different regions, CxFV ecology and the influence of co-circulation of arboviruses remains poorly understood. Therefore, our primary goals are to observe the occurrence of CxFV infection in mosquitoes trapped in an urban area of Rio de Janeiro, Brazil, characterize the virus circulating, and provide isolates. A prospective study was carried out for eight months on the campus of the Federal University of Rio de Janeiro, trapping adult mosquitoes. The CxFV minimum infection rates were determined in this period, and the virus isolation process is fully described. Samples from this study were grouped into genotype 2, along with CxFV sequences from Latin America and Africa.

## 1. Introduction

The Flavivirus genus groups a wide range of species capable of infecting vertebrates and invertebrates, both terrestrial and aquatic [1]. The diversity of hosts and ecological cycles increase the complexity of this genus. The flavivirus genome is composed of a single positive-stranded RNA, usually containing a single ORF translated into a polyprotein. This polyprotein undergoes cleavage giving rise to mature viral proteins, divided into three structural proteins, capsid (C), envelope (E), pre-membrane/membrane (prM/M) proteins; and seven non-structural proteins (NS1-S2A-NS2B-NS3-NS4A-2K-NS4B-NS5). The ORF is flanked by 5′ and 3′ untranslated regions (UTRs) [2,3,4].

According to phylogenetic analyses, the flavivirus genomes cluster into three main branches; the first one containing viruses that infect vertebrates, also called arboviruses; the second called arbovirus-affiliated insect-specific flaviviruses or dual-host insect-specific flaviviruses (dISF), that preserve genomic similarity with arboviruses, but its replication is apparently restricted to invertebrates and insect-specific classical flaviviruses (ISF), with infection restricted to invertebrates [1]. *Culex Flavivirus* (CxFV) is a classical insect-specific virus, which has aroused interest after the first indication that colonies of *Culex Pipiens* mosquitoes naturally infected by CxFV presented a delayed dissemination of West Nile virus in a vector competence study [5]. The first detection occurred in *Culex Pipiens* mosquitoes captured between 2003 and 2004 in Japan and Indonesia [6]. This virus was then isolated from different species of mosquitoes of the *Culex* genus in countries in South America (Brazil and Argentina), Central America (Guatemala and Trinidad), North America (USA), and Africa (Uganda) [7,8,9,10,11]. To date, two genotypes have been identified for this species, genotype I that groups together isolates of USA and Japan and genotype II with isolates of South America, Central America, and Africa [10,12,13,14]. 

The first report of CxFV detection in Brazil was in 2012, in *Culex spp*. mosquitoes, collected between July 2007 and January 2008 in the city of São José do Rio Preto, State of São Paulo [7]. Since then, infected mosquitoes have been detected in different regions of Brazil. In 2013, *Culex quinquefasciatus* Say, 1823 mosquitoes positive for CxFV were detected in the Midwest in Cuiabá [15]. Between 2011 and 2013, *Culex chidesteri* Dyar, 1921 infected were found in the Northeast, in the State of Rio Grande do Norte [16]. A study published in 2018 reported a high prevalence of insect-specific Flaviviruses in *Culex* spp. captured in the state of Espírito Santo, Southeast, in 2016 and in the state of Paraná, southern Brazil, in 2017 [17]. To date, there are no descriptions of detection of CxFV in Rio de Janeiro.

The data regarding the ecology of CxFV and transmission dynamics are scarce. Laboratory experiments using established colonies of naturally-infected mosquitoes pointed out vertical transmission as the primary mode of maintenance of the virus in the *Culex pipiens* species. The transmission mode was observed on a smaller scale, and there was no record of horizontal transmission (Figure 1). However, persistently-infected mosquitos experienced a delayed dissemination process of West Nile virus compared with non-infected mosquitoes. This evidence reinforces the hypothesis of interference of CxFV on vector competence and the enzootic transmission of arboviruses [18].

Therefore, providing CxFV isolates has become highly relevant to input studies of superinfection and increases understanding of vector competence’s stochastic phenomenon. Since several arboviruses such as Dengue, Zika and Chikungunya are endemic in Rio de Janeiro our primary goals in this study are to observe the occurrence of CxFV infection mosquitoes trapped in an urban area of Rio de Janeiro, Brazil, and to characterize the virus circulation, and provide isolates.

## 2. Materials and Methods

### 2.1. Sample Collection

A prospective study of CxFV in mosquitoes was carried out for eight months on the campus of the Federal University of Rio de Janeiro between June 2019 and March 2020. The campus is located on an island (Ilha do Fundão) in Guanabara Bay near the mainland. The climate is tropical with an average annual temperature of 28 °C, relative humidity of up to 84%, and a rainy period between (May and October). The collection was carried out using BG sentinel traps (BGS). Each BGS trap was baited with a BG-Lure (Biogents AG) and placed in a wooded area with a circulation of people (22°50′32.4″ S 43°14′03.5″ W) for 24 h (Figure 2). The captured mosquitoes were transported to the laboratory, counted, sexed, and identified at the genus level using the taxonomic key proposed by Consoli and Lourenço-de-Oliveira et al. [19]. The capture was authorized by Sisbio license number 54192. To investigate the viral infection, mosquitoes were grouped into pools of a maximum of 250 individuals according to the day of capture, sex, and genus.

### 2.2. CxFV Detection by qRT-PCR

Mosquitoes were homogenized in 2.0 mL tubes with 1000 µL of Dulbecco’s Modified Eagle’s Medium (DMEM) supplemented with 3% fetal calf serum, 2.5 µg/mL amphotericin B, 500 U/mL penicillin, 100 µg/mL streptomycin and Zirconia beads (2.0 mm, Cat. No. 11079124zx Biospec products). Then, the mosquitoes were macerated by vortexing, and centrifuged at 4500 rpm (or 3000× *g*) for 15 min at 4 °C. The supernatant was removed and used for RNA extraction and isolation. The QIAamp Viral RNA Kit (QIAGEN, Inc., Valencia, CA, USA) was used to extract RNA according to the manufacturer’s instructions, from 140 µL of the supernatant obtained after maceration. The qRT-PCR described by [20] was used to screen for CxFV. Positive samples in qRT-PCR were amplified by conventional RT-PCR described by [21] using primers FU2 and cFD3 targeting the NS5 region. In addition, we screened the samples for the presence of all endemic arbovirus in Rio de Janeiro The following RT-qPCR protocols were used for ZIKV [22], CHIKV [23], DENV [24] and YF [25]. The engorged females were tested individually.

### 2.3. Cell Culture

To perform virus isolation, monolayers of *Aedes albopictus* Skuse C6/36 cells were cultivated at 28 °C in Leibovitz’15 (L15) medium supplemented with 5% fetal bovine serum, 5% tryptose phosphate broth, penicillin G 100 IU/mL and streptomycin 100 g/mL.

### 2.4. Virus Isolation

Samples with C t values below 27 (samples 3893: Ct = 26.44; 3897: Ct = 26.77; 3918: Ct = 24.60; 3929: Ct = 26.41) were selected for isolation of CxFV. C6/36 monolayers with 90% confluence were prepared in culture tubes with a 10 cm^2^ growth area. The initial inoculum of 200 µL of supernatant, filtered through 0.45 µm membranes, was used. After 1 h incubation in rocker platform (Vari Mix Platform Rocker ThermoFisher©) at minimal speed for viral adsorption, the inoculum was removed from the C6/36 cell monolayer, and a fresh L15 culture medium containing 5% FBS was added. Cells were incubated at 28 °C for up to 7 days. Cell culture tubes were evaluated daily for cytopathic effect (CPE). Regardless of the presence of CPE, after seven days of infection, the supernatant was collected, centrifuged for 10 min at 5000 rpm, and 200 µL were used as inoculum for the second passage in C6/36 cells. Isolation was confirmed by qRT-PCR; sample 3929 had the lowest Ct value 16.47 and was chosen to produce a large number of viruses. To achieve a high viral load, sequential passages were performed. Aliquots from both passages were collected for RT-PCR and plaque-forming units (PFU) titration.

### 2.5. Plaque-Forming Units

For viral titration by plaque-forming units (PFU), monolayers of C6/36 cells were grown in 6-well plates to 80% confluence. The supernatant of the sixth passage of the sample, named 3929 (with higher viral load by qRT-PCR), was serially diluted in base 10 to 10^6^ using L15 medium without FBS supplementation. An inoculum of 700 µL of the dilutions 10^2^ to 10^6^ was used per well. Virus adsorption was carried out for 1 h in a rocker platform (Vari Mix Platform Rocker ThermoFisher©) at minimal speed for viral adsorption. After the adsorption period, 2 mL of a 1:1 mixture of 2% Carboxymethylcellulose (CMC-Sigma) and 2× concentrated L15 medium supplemented with 10% FBS was added to each well. The plate was incubated at 28 °C for 10 days. After incubation, cells were fixed with 10% formaldehyde for at least 24 h and stained with 2% crystal violet. Plaque-forming units were counted for viral titer determination.

### 2.6. Sequencing and Phylogenetic Analysis

Two fragments of CxFV genome were sequenced: a part of NS5 gene, using the FU2 and cFD3 primers described by Kuno et al. (1998) [21]; and the coding regions of proteins C, prM/M and part of protein E using primers A-1F GGATGACGTCCAGCAACTCATCAGTGA and A-1R CGCACAAACAATCCTTCGTGGTATTTG described by Machado (2016) [26]. The amplicons generated were sequenced on the ABI 3730 Genetic Analyzer apparatus (Applied Biosystems, Foster City, CA, USA) using the Big Dye Terminator V3.1 kit sold by Applied Biosystem, following the manufacturer’s protocol. Phylogenetic analysis was carried out by collecting sequences from the NS5 region of CxFV and from other insect-specific viruses available in the GenBank database. In total, 5 CxFV sequences and 11 sequences from other insect-specific viruses were obtained. The sequences were aligned, edited, and assembled using the BioEdit Sequence Alignment Editor, version 7.0.5.3 [27]. The best nucleotide substitution model for calculating evolutionary distances was evaluated using the MEGA v.7.0.2 program, the models that best fit for each region are described in the figure note. The phylogenetic trees were constructed using the Maximum Likelihood and Neighbor Joining methods. The statistical significance of the different phylogenies was obtained by the Maximum Likelihood (ML) method using 1000 bootstrap replicates [28]. Only percentages over 50 are shown at the node. Evolutionary analyses were conducted in MEGA 7.0.2.

### 2.7. Light Microscopy of CxFV Infected Cells

Mock and infected monolayers were observed in an Axio Observer Z1 Zeiss microscope equipped with a HXP light source, using the Nomarski differential contrast system.

### 2.8. Confocal Microscopy of CxFV Infected Cells

For immunofluorescence microscopy, C6/36 persistently infected cells were seeded on round coverslips and fixed with 4% formaldehyde in phosphate-buffered saline (PBS), pH 7.2, for 20 min. The samples were permeabilized with 0.1% Triton X-100 in PBS for 10 min at room temperature. Pre-incubation was performed with 50 mM ammonium chloride and 3% BSA in PBS, pH 8.0, for 45 min to block the free aldehyde groups. The samples were then incubated with a primary anti-flavivirus antibody 4G2 hybridoma, gently provided by Prof. Luciana Arruda and Prof. Fábio Gomes (UFRJ) at a 1:100 dilution for 1 h, rinsed, and incubated with a 1:400 dilution of the secondary goat anti-human IgG antibody conjugated to AlexaFluor 488 (Invitrogen) at room temperature for 1 h. Actin was stained with actin red (Invitrogen) for 20 min in the dark. After rinsing with PBS and mounting with prolong antifade (Vector Labs, Burlingame, CA, USA), the slides were visualized using a Zeiss Elyra PS.1, using Confocal mode.

### 2.9. Statistical Analysis

Data from mosquito collection were tabulated into a spreadsheet and aggregated by each collection day. Exploratory data analysis was performed by grouping data by species and ingurgitation status (only for females). Descriptive statistics (mean, median, standard deviation, minimum and maximum values) were calculated to analyze entomological data. Monthly collection data were standardized by calculating the ratio between the total number of individuals captured per month and the number of visits performed to assess the distribution of individuals over the months of capture for each species. The minimum infection rate (MIR) was calculated as the number of positive pools divided by the total number of mosquitoes tested and expressed as a percentage. Finally, we were also interested in analyzing the temporal aspect of MIR during the study period. In this case, MIR was calculated for each month of study. All analyzes were performed in R and RStudio software [29,30].

## 3. Results

### 3.1. Mosquitoes Study Population

The present study reported the first identification of CxFV in naturally-infected mosquitoes of the *Culex* genus in Rio de Janeiro, southeastern Brazil. During the capture period of this study, a total of 760 mosquitoes were collected. Of this total, 401 belonged to the *Culex* genus (51.1%) and 359 to the *Aedes* genus (48.9%). The population distribution according to sex showed more females of the *Aedes* genus captured than males. The average capture of females per day was 3.6, and 2.52 for males. In the *Culex* genus, the opposite was observed; average capture of females per day was 2.33, and 4.38 for males (Table 1).

Data were normalized by the ratio between the total number of individuals captured per month and the number of visits performed to assess the distribution of individuals over the months of capture. The distribution of *Aedes* mosquitoes was less noisy, i.e., the variation between the highest and lowest number of captured mosquitoes was smaller than that presented for the *Culex* population. The genus *Aedes* showed a downward trend from the beginning of the series until October, with an increase until February and a subsequent decrease. For *Culex*, the distribution is noisier, with a valley from June to September, a ramp in October, and a peak in November, with a subsequent fluctuation in density. The peak of the *Aedes* collection occurred in July 2019, with 91 individuals collected, while the peak of *Culex* occurred in November, with 131 individuals collected. The largest number of mosquitoes acquired in a single collection was 13 individuals for both species (Figure 3).

### 3.2. Culex Flavivirus Minimum Infection Rate (MIR)

The pools of mosquitoes were tested for the presence of CxFV genomic material. All *Aedes* spp. were negative. Of the total of 134 pools from *Culex* spp. (401 individuals), 31 were positive with an average of Ct 26.52, giving a minimum infection rate of 7.73% (MIR, calculated as the number of positive pools divided by the total number of mosquitoes tested and expressed as a percentage). The highest number of positive pools was found in July, with eight infected pools from 53 captured individuals (MIR of 15.09), followed by August with seven pools and September with five pools (from a total of 35 mosquitoes collected in August and 29 in September, MIR of 20 and 17.24, respectively) (Table 2). The average number of positive samples for viral RNA was 3.875 per month and a median of 3.5. It was also possible to analyze that of the total of positive *Culex* pools (31), only 11 (35.4%) were female mosquitoes (Figure 4).

All samples were screened for other arboviruses (Dengue virus all serotypes, Zika virus, Chikungunya virus and Yellow fever virus) using qRT-PCR as described in Section 2.2. However, none were positive.

### 3.3. CxFV Isolation, PFU Titration, and Microscopy

A sample of CxFV was isolated from C6/36 cells as described in Section 2. The first passages did not result in CPE, and a decrease in the growth rate of the cell monolayer was observed. Among the 4 samples selected for isolation, sample 3929 showed the highest viral load by qRT-PCR, being then chosen to produce viral stock. After the fourth pass of sample 3929, a discrete CPE was detected (Figure 5A–D). In the sixth passage, it was possible to titrate the CxFV by PFU. The titer of 2.8 × 10^6^ was obtained following the methodology described in Section 2.

Immunofluorescence assay (Figure 5E,F) corroborated the previous data and showed the localization of viral particles within infected cells.

### 3.4. Phylogenetic Analysis of CxFV

The identity of the virus detected in this study was proved by sequencing 533 bp fragments of the gene encoding the NS5 protein. The five samples sequenced showed between 100 and 99.2% similarity to each other. Compared with other described CxFVs, the nucleotide similarity was close to 98% with the strain detected in Kenya in 2012 (GenBank LC348554), thus confirming the identity of the identified species.

The phylogenetic analysis of CxFV from Rio de Janeiro was performed with 533bp of the gene encoding the NS5 protein and 1243bp of the region encoding the structural proteins. The NS5 region was aligned with 12 CxFV sequences and 11 sequences from other insect-specific viruses. The tree obtained showed two main monophyletic branches: genotype I and genotype II. Genotype I comprises isolates from the USA, Japan, Thailand, China, and Vietnam, and genotype II consists of isolates from Brazil, Mexico, and Uganda. Samples from this study were grouped into genotype II, along with CxFv sequences from Latin America and Africa (Figure 6a). Phylogenetic analysis of the structural protein genes corroborated the findings of the NS5 region. The CxFV isolated in Rio de Janeiro remained grouped with genotype II (Figure 6b). Unfortunately, the number of nucleotide sequences presenting the structural region or complete genome available in the databases is limited. The sequences from this study have been deposited and are available from GenBank under accession numbers: MT683843, MT683844, MT683845, MT683846, and MT683847.

## 4. Discussion

CxFV had been described in many regions, suggesting a widespread distribution in *Culex* species. Although, few studies followed the infection rate in the field population, most of them were restricted to the point detection and genome characterization of this virus. No description of CxFV detection and isolation from field mosquitos of Rio de Janeiro, Brazil, had been reported before our work.

In the present study, the occurrence of CxFV infection in the field population was followed over eight months. The continuous trapping of adult mosquitoes was performed with BGS traps. Although BGS traps were designed to monitor mosquitoes of the *Aedes* genus, they are also efficient for capturing the *Culex* species [31,32]. These traps use substances with odors similar to human pheromones, so they are specific to host-seeking for blood-feeding. Therefore, *Ae. aegypti* females were captured more than males, reaffirming the efficiency of this trap for monitoring Aedes. However, a large number of individuals of the genus *Culex*, males and females, were trapped. Maciel-de-Freitas et al. (2006) [33] obtained similar data during the BGS validation study conducted in the Tubiacanga neighborhood, a region located approximately 12 km away from the collection point of our research. Moreover, according to the observations of Maciel-de-Freitas et al. [33], we identified a greater natural abundance of the genus *Culex*.

We confirmed the presence of CxFV RNA in male and female mosquitoes. Males presented higher MIR than females, 8.66 and 6.66, respectively. However, an accurate estimation of prevalence by sex depends on more extensive sampling. Meanwhile, the large number of male mosquitoes positive for CxFV RNA contributes to the hypothesis that vertical transmission is essential for maintaining this virus in the mosquito population [11,18,34].

In addition, these results suggested the veracity of the seasonality hypothesis for CxFV infection, since the curves produced by positive pools and the number of mosquitoes trapped differ over the months. August and September showed the highest MIRs and the lowest amounts of mosquitoes caught (Figure 4). The decrease in mosquitoes infected with CxFV in the summer was also observed in the northern hemisphere, suggesting that temperature may influence the prevalence of CxFV [5]. Correlation analyses between phylogeny and climate variables revealed a strong association between CxFV genotypes and weather. Genotype I consists of viruses from temperate zones, while genotype II, which also includes the sequences from this study, consists of viruses from tropical regions [35]. CxFV is widely disseminated and has been found in several countries in Asia, South America, Central America, North America, and Africa. Genetically, CxFV is considered stable, and the phylogenic trees cluster geographically, even though isolation occurs from different hosts, suggesting a possible transmission between sympatric species [36]. The movement between continents could have occurred as described by other arboviruses [12]. In Brazil, CxFV was reported in four geographic regions (Midwest, Northeast, Southeast, and South); however, the number and length of genomic sequences available in data banks limit robust phylogenetic analyses and dissemination studies.

In this study, it was possible to isolate and titrate a sample of CxFV in C6/36 cells by PFU. As previously reported by other authors, we did not observe a CPE in the first passages in c6/36 cells [6,8,12]. The production of characteristic CPE and formation of syncytia in C6/36 cells were described only for a few isolates of CxFV from Mexico [12]. However, after the fourth passage of the isolate 3929 from Rio de Janeiro, it was possible to observe a discrete CPE and the formation of Syncytia (Figure 4); after the sixth passage, it was possible to perform titration by PFU. Considering that the cells used for isolation came from another vector, the consecutive passages may have increased the fitness of the virus in this system.

Despite advances in the discovery of ISVs in arbovirus vector populations, the influence of co-circulation of these viruses remains poorly understood. Newman et al. (2011) [5] found in nature a positive association between the mosquitoes of the genus *Culex* infected with West Nile virus (WNV) and CxFV in Chicago, from 2011 to 2012. This positive association agrees with Kent’s results in 2010 that did not observe interference of West Nile replication when CxFV previously infected C6/36 cells. In the same study, Kent demonstrated no significant interference in the vector competence for WNV when CxFV previously infected *Cx. quinquefasciatus*. However, this competence increased when mosquitoes were simultaneously infected with the two viruses [37].

On the contrary, Bolling and colleagues in 2012 [15] described a delay in the spread of WNV when *Cx. pipiens* mosquitoes were superinfected, which suggests the possibility of impacting enzootic transmission in co-circulating regions. This result reflects the variability of CxFV phenotypes in different mosquito species, which poses an enormous challenge for understanding vector competence and the dynamics of arboviruses emergence. 

The isolation of CxFV with high viral load will allow in vivo and in vitro co-infection studies, morphogenesis analysis by electron microscopy, and evaluation of the immune response by transcriptome. Isolation and characterization of different CxFV strains will advance superinfection studies and flavivirus evolution.

## 5. Conclusions

This study describes the natural occurrence of CxFV in the mosquito population of the urban area of Rio de Janeiro for the first time. The infection of males and females was confirmed, and the seasonality of occurrence was observed. The virus detected belongs to genotype II, which corroborates the isolates previously reported in Latin America. The isolation of this virus will allow in vivo and in vitro superinfection experiments and contribute to the studies of flavivirus evolution.

## Figures and Tables

**Figure 1 insects-13-00477-f001:**
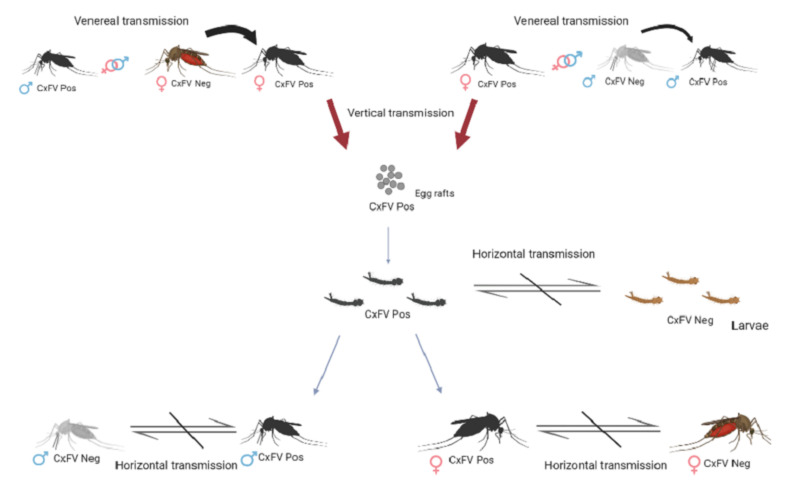
Sscheme summarizes the results of transmission routes obtained wcheme of CxFV transmission. The ith a laboratory colony of naturally-infected *Cx. pipiens*. The width of the arrow represents the probability of virus transmission. The crossed arrows represent no evidence of virus transmission [18].

**Figure 2 insects-13-00477-f002:**
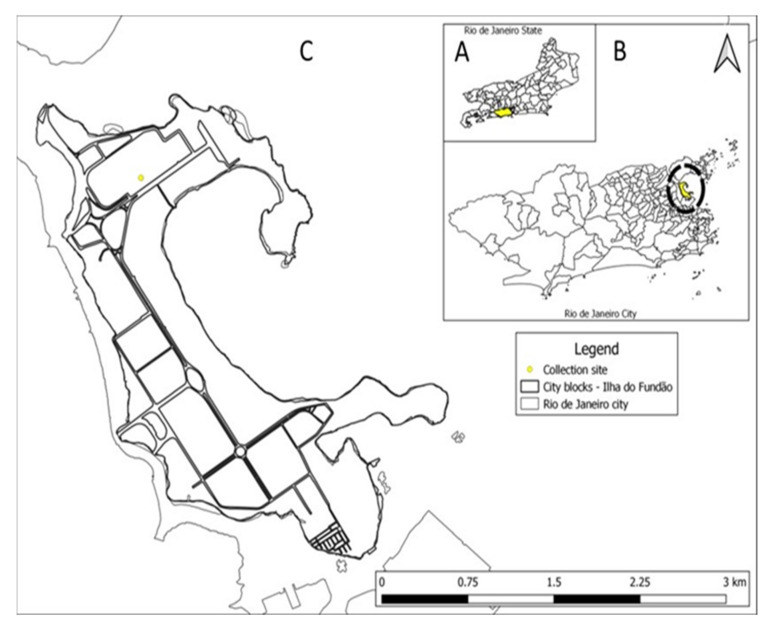
The geographic area of mosquito trapping. A—State of Rio de Janeiro; B—Rio de Janeiro City; C—University Island.

**Figure 3 insects-13-00477-f003:**
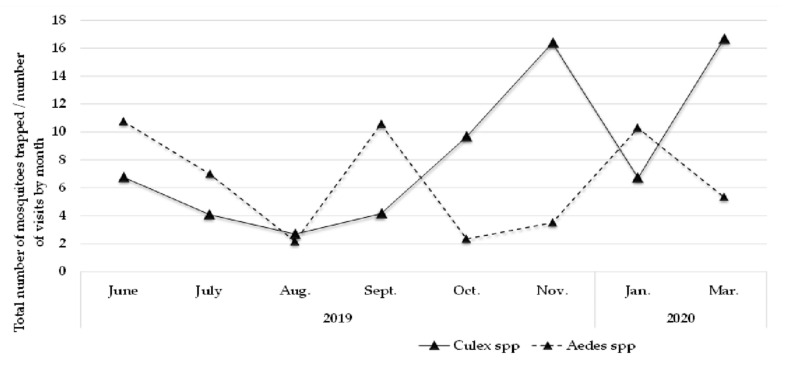
Graphic distribution of mosquitoes collected by months. The total number of individuals collected in each month was divided by the total number of visits carried out. Peaks for the Aedes genus were observed in June and September of 2019, and also in January of 2020; while peaks for the *Culex* genus were observed in November of 2019 and March of 2020, during the summer season.

**Figure 4 insects-13-00477-f004:**
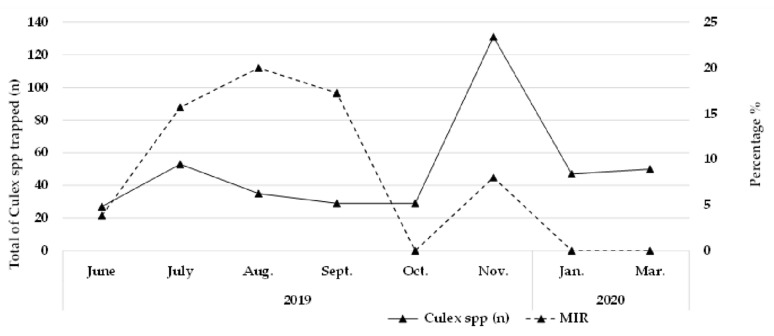
Time series of *Culex* spp. trapped and CxFV MIR per month of the study. The graph Scheme 2020. no *Culex* positive pools were observed.

**Figure 5 insects-13-00477-f005:**
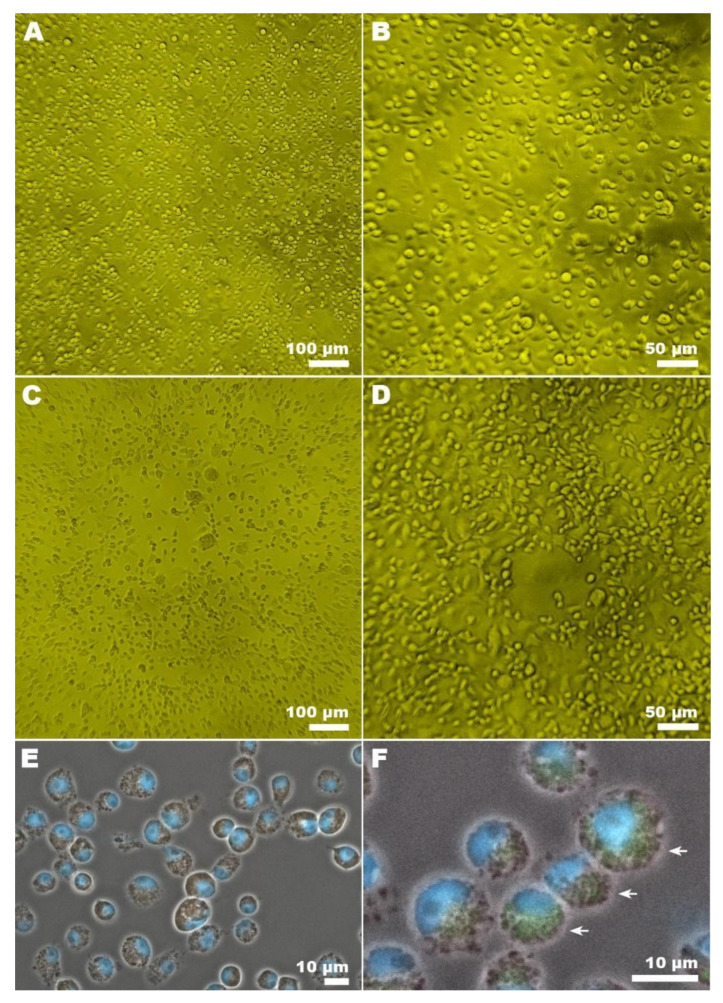
Light microscopy analysis of CxFV infection in C6/36 cells. Low (10×) and high magnification (20×) images of mock (**A**,**B**) and infected cells (**C**,**D**). CPE was observed (**C**,**D**) in CxFV infected cells. (E) Immunofluorescence showed no labelling in mock-infected cells, while CxFV was labelled (green color, arrows) in infected cells (**F**).

**Figure 6 insects-13-00477-f006:**
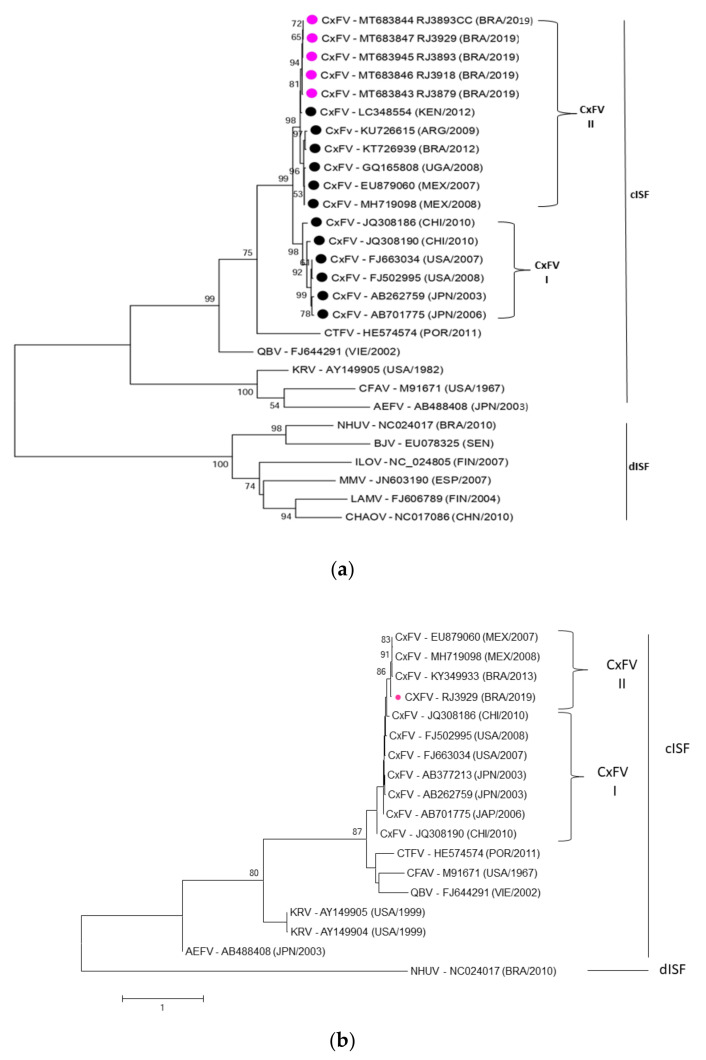
Genetic relationship of CxFV with insect-specific flaviviruses. (a) The phylogenetic tree was constructed for the NS5 region (533pb). The evolutionary history was inferred by using the Maximum Likelihood method based on the General Time Reversible model. The nucleotide substitution model that best fit for the NS5 region is K2 + G + I. A discrete Gamma distribution was used to model evolutionary rate differences among sites by assuming that a certain fraction of sites are evolutionarily conserved. The branch support was assessed by the bootstrap method with 1000 replicates. Only percentages over 50 are shown at the node. Pink dots—sequences from this work; black dots—CxFV sequences. (**b**) The phylogenetic tree was constructed for the region encoding the structural proteins C, prM/M, and part of protein E (1243pb). The evolutionary history was inferred using the Neighbor-Joining method and the evolutionary distances were computed using the Kimura 2-parameter. The rate variation among sites was modeled with a gamma distribution (shape parameter = 1). The branch support was assessed by the bootstrap method with 1000 replicates. Only percentages over 50 are shown at the node. Evolutionary analyses were conducted in MEGA 7. Red dots—sequence from this work. CTFV: Culex theileri flavivirus; QBV: Quang Binh virus; KRV: Kamiti River Virus; CFAV: Flavivirus cell fusing agent; AEFV: Aedes Flavivirus; NHUV: Nhumirim Virus; BJV: Barkedji virus; ILOV: Ilomantsi virus; MMV: Marisma mosquito virus; LAMV: Lammi virus; CHAOV: Chaoyang virus.

**Table 1 insects-13-00477-t001:** Mosquitoes collected in the Center of Sciences and Health (from the Federal University of Rio de Janeiro), broken down by species, sex, and engorgement status.

Specie	Sex	n/pool	Mean	Median	Min.	Max.	Standard Deviation	Percent
*Aedes* spp.	Female	209/69	3.60	2	0	18	4.43	27.50
	Female-eng ^1^	4	0.069	0	0	2	0.32	0.53
	Male	146/49	2.53	1	0	19	3.96	19.21
*Culex* spp.	Female	135/50	2.33	1	0	10	2.74	17.76
	Female-eng ^1^	12	0.21	0	0	2	0.52	1.58
	Male	254/72	4.38	3	0	15	3.87	33.42

^1^ Female-eng. Engorged female.

**Table 2 insects-13-00477-t002:** Total of *Culex* mosquitoes, viral RNA positive pools, and Minimum infection rate (MIR), broken down by year and month of collection.

Year	Month	Males *n*	Positive pools	Females *n*	Positive pools	MIR *
2019	June	16	1	11	1	3.7037
	July	39	7	14	1	15.0943
	Aug.	27	6	8	1	20
	Sept.	21	2	8	3	17.2414
	Oct.	12	0	17	0	0.0000
	Nov.	79	6	52	4	7.63359
2020	Jan.	28	0	19	0	0.000000
	Mar.	32	0	18	0	0.000000
Total		254	22	147	9	7,730,673

* MIR (number of positive pools divided by the total number of mosquitoes tested and expressed as a percentage).

## Data Availability

Data are contained within the article.

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
