# Peer review of "Culex Flavivirus Isolation from Naturally Infected Mosquitoes Trapped at Rio de Janeiro City, Brazil"

_insects, 2022, doi:10.3390/insects13050477_

Round 1

Reviewer 1 Report

In this study, Amaral et al. observed the occurrence of Culex Flavivirus (CxFV) infection mosquitoes trapped in an urban area of Rio de Janeiro, Brazil, characterized the virus circulating, and provided isolates. Your manuscript is interesting especially because CxFV has been showing interference on vector competence and the enzootic transmission of arboviruses of medical importance. However, I suggest some corrections in this manuscript for better clarification and originality of the work.

Major revision:

  1. Although this study is pioneer in the detection and isolation of Culex Flavivirus from field mosquitoes in Rio de Janeiro, as reported by the authors, in the literature there are some published studies showing detection, identification and isolation of this virus in other regions of Brazil (a. Med Vet Entomol. 2019 Sep;33(3):397-406. https://doi.org/10.1111/mve.12374. b. Entomological Communications, 3, ec03018. https://doi.org/10.37486/2675-1305.ec03018. c. Acta Trop. 2016 May; 157:73-83. https://doi.org/10.1016/j.actatropica.2016.01.026. d. Virology. 2019 Jan 15; 527:98-106. https://doi.org/10.1016/j.virol.2018.11.008). For a better contextualization of this study, in the introduction section, I would like the authors write a paragraph exploring this information.

  1. Still in the introduction section, I don't see the need for figure 1. I recommend that authors remove.

  1. In the Material and Methods section, item 2.1, please inform the period in which the collections were carried out (months and year). Although it is mentioned in the results, I recommend bringing this information in the methodology.

  1. In fact, the collections were carried out in eight months and not in ten as mentioned in the Material and Methods section, item 2.1. Informations about mosquito sample collections are important and can contribute to the reproducibility of the study. I strongly advise the author to write in more detail and clarity.

  1. In item 2.5 of the Materials and Methods section where the authors describe the titration by Plaque Forming Unit (P.F.U) only the supernatant from the sixth passage of the sample named 3929 is mentioned. However, in the previous item are mentions 3 other samples with Ct value below 27 that were used for isolation (3893, 3897 and 3918). Why the titration of these three samples were not performed? Please, clarify. I advise authors to use the correct nomenclature which is Ct (Cycle threshold) instead of "ct" in the text.

  1. I strongly recommend that the authors include an item “Statistical analysis” in the Materials and Methods section. In this item, inform all calculations that were performed and are being shown in the Result section (mean, median, min., max., standard deviation percent, minimum infection rate). If possible, inform which program (software) and version was used.

  1. In Material and Methods, item 2.6, a better description of the parameters of the phylogenetic tree constructed was lacking. Maybe mention in this item what was described in the Results.

  1. The authors mention in the item 3.1 of the Result section that a total of 760 mosquitoes were collected. Of this total, 401 belonged to the Culex genus (51.1%) and 359 to the Aedes genus (48.9%). It is unclear how many pools of each mosquito genus were formed and analyzed. Please, clarify.

  1. Why did the authors not research the presence of other viruses in the pools of the Culex genus and Aedes genus?

  1. Again, in item 3.3 of the Results section, the authors base themselves on the sample titration 3929. And the other samples that also were used in the isolation? Please, provide details of these samples.

  1. The samples with Ct values below 27 were selected for isolation of CxFV. However, and the other positive samples with Ct above 27? I recommend that authors indicate the average of the Cts of these samples.

  1. The Aedes spp. pools were not analyzed?

  1. The authors mention that they observed a total of 31 pools of Culex spp. positive for CxFV. This is the total number of pools of Culex spp. or a part of a larger set? It is not clear in the results section, please clarify.

  1. In the Discussion section, I would have wished to see more information on circulation

of CxFV.

  1. I think the isolation of the virus described in the results was poorly discussed. Please improve this aspect.

  1. Please, include the limitations and future perspectives of your study in the Discussion section.

Author Response

Major revision:

  1. Although this study is pioneer in the detection and isolation of Culex Flavivirus from field mosquitoes in Rio de Janeiro, as reported by the authors, in the literature there are some published studies showing detection, identification and isolation of this virus in other regions of Brazil (a. Med Vet Entomol. 2019 Sep;33(3):397-406. https://doi.org/10.1111/mve.12374. b. Entomological Communications, 3, ec03018. https://doi.org/10.37486/2675-1305.ec03018. c. Acta Trop. 2016 May; 157:73-83. https://doi.org/10.1016/j.actatropica.2016.01.026. d. Virology. 2019 Jan 15; 527:98-106. https://doi.org/10.1016/j.virol.2018.11.008). For a better contextualization of this study, in the introduction section, I would like the authors write a paragraph exploring this information.

A: We thank you for the references. We included the following paragraph in the Introduction section, covering these suggested references.

“The first report of CxFV detection in Brazil was in 2012, in Culex spp. mosquitoes, collected between July 2007 and January 2008 in the city of São José do Rio Preto, State of São Paulo [7]. Since then, infected mosquitoes have been detected in different regions of Brazil. In 2013, Culex quinquefasciatus Say, 1823 mosquitoes positive for CxFV were detected in the Midwest in Cuiabá [15]. Between 2011 and 2013, Culex chidesteri Dyar, 1921 infected mosquitoes were found in the Northeast, in the State of Rio Grande do Norte [16]. A study published in 2018 reported a high prevalence of insect-specific Flaviviruses in Culex sp. captured in the state of Espírito Santo, Southeast of Brazil, in 2016; and in the state of Paraná, southern Brazil, in 2017 [17]. To date, there is no description of CxFV detection in Rio de Janeiro.”

  1. Still in the introduction section, I don't see the need for figure 1. I recommend that authors remove.

A. The figure was constructed to graphically summarize the evidence found so far on how the virus is transmitted in mosquitoes, facilitating the understanding of readers who are not familiar with the ecology of specific insect viruses. However, if the reviewer still considers this unnecessary, we can remove it.

  1. In the Material and Methods section, item 2.1, please inform the period in which the collections were carried out (months and year). Although it is mentioned in the results, I recommend bringing this information in the methodology.

A. Thank you for this observation. The months and years of mosquito collection were included in the Material and methods as follows:

  1. Materials and Methods

2.1. Sample collection

  “A prospective study of CxFV in mosquitoes was carried out for eight months on the campus of the Federal University of Rio de Janeiro between June 2019 up to March 2020.”

  1. In fact, the collections were carried out in eight months and not in ten as mentioned in the Material and Methods section, item 2.1. Informations about mosquito sample collections are important and can contribute to the reproducibility of the study. I strongly advise the author to write in more detail and clarity.

A. We thank you for the note, the period of collection was corrected all over the manuscript.

  1. In item 2.5 of the Materials and Methods section where the authors describe the titration by Plaque Forming Unit (P.F.U) only the supernatant from the sixth passage of the sample named 3929 is mentioned. However, in the previous item are mentions 3 other samples with Ct value below 27 that were used for isolation (3893, 3897 and 3918). Why the titration of these three samples were not performed? Please, clarify. I advise authors to use the correct nomenclature which is Ct (Cycle threshold) instead of "ct" in the text.

A: After the two initial passages, all samples showed amplification by qRT-PCR (3893, 3897, 3918, and 3929), indicating isolation. However, passage 3929 showed the highest viral load. For this reason, this passage was chosen to produce a viral stock and titer increase. Therefore, we performed consecutive passages only on sample 3929 until observing the cytopathic effect (sixth passage). The sixth passage was titrated. The other samples were stored in the second passage for future studies. This information was included in Materials and Methods as follows:

2.4. Virus Isolation

Regardless of the presence of CPE, after seven days of infection, the supernatant was collected, centrifuged for 10 min at 5,000 rpm, and 200 µl were used as inoculum for the second passage in C6/36 cells. Isolation was confirmed by qRT-PCR; sample 3929 had the lowest Ct value 16.47 and was chosen to produce a large number of viruses. To achieve a high viral load, sequential passages were performed. Aliquots from both passages were collected for RT-PCR and plaque-forming units (PFU) titration.

2.5. Plaque-forming units

“For viral titration by Plaque-forming units (PFU), monolayers of C6/36 cells were grown in 6-well plates to 80% confluence. The supernatant of the sixth passage of the sample, named 3929 (with higher viral load by qRT-PCR), was serially diluted in base 10 to 106 using L15 medium without FBS supplementation”

We also thank you for the correction concerning the Cycle threshold nomenclature, which was) was corrected.

  1. I strongly recommend that the authors include an item “Statistical analysis” in the Materials and Methods section. In this item, inform all calculations that were performed and are being shown in the Result section (mean, median, min., max., standard deviation percent, minimum infection rate). If possible, inform which program (software) and version was used.

A: We thank you for the suggestion. We have attended it by adding the following text at the end of the Materials and Methods section:

2.9. Statistical Analysis

“Data from mosquito collection were tabulated into a spreadsheet and aggregated by each collection day. Exploratory data analysis was performed by grouping data by species and ingurgitation status (only for females). Descriptive statistics (mean, median, standard deviation, minimum and maximum values) were calculated to analyze entomological data. Monthly collection data was standardized by calculating the ratio between the total number of individuals captured per month and the number of visits performed to assess the distribution of individuals over the months of capture for each species. The minimum infection rate (MIR) was calculated as the number of positive pools divided by the total number of mosquitoes tested and expressed as a percentage. Finally, we also were interested in analyzing the temporal aspect of MIR during the study period. In this case, MIR was calculated for each month of study. All analyzes were performed in R and RStudio softwares [29-30].”

  1. In Material and Methods, item 2.6, a better description of the parameters of the phylogenetic tree constructed was lacking. Maybe mention in this item what was described in the Results.

A: We agree with Reviewer #1. We have now added in Material and Methods the common information for the construction of trees as follows:

2.6. Sequencing and phylogenetic analysis

“The sequences were aligned, edited, and assembled using the BioEdit Sequence Alignment Editor, version 7.0.5.3 [27]. The best nucleotide substitution model for calculating evolutionary distances was evaluated using the MEGA v.7.0.2 program, the models that best fit for each region are described in the figure note. The phylogenetic trees were constructed using the Maximum Likelihood and Neighboor Joining methods. The statistical significance of the different phylogenies was obtained by the Maximum Likelihood (ML) method using 1,000 bootstrap replicates     [28].  Only percentages over 50 are shown at the node. Evolutionary analyses were conducted in MEGA 7.0.2.

  1. The authors mention in the item 3.1 of the Result section that a total of 760 mosquitoes were collected. Of this total, 401 belonged to the Culex genus (51.1%) and 359 to the Aedes genus (48.9%). It is unclear how many pools of each mosquito genus were formed and analyzed. Please, clarify.

A: In fact, this information was absent. The number of pools formed by genus was now included in table1.

  1. Why did the authors not research the presence of other viruses in the pools of the Culex genus and Aedes genus?

A: The authors performed the research on other arboviruses; however, no positives were found. This information was now included at the end of item 3.2, as follows:

3.2. Culex Flavivirus minimum infection rate (MIR)

" The pools of mosquitoes were tested for the presence of CxFV genomic material. All Aedes spp were negative.  Of the total of 134 pools from Culex spp. (401 individuals), 31 were positive with an average of Ct 26.52, giving a minimum infection rate of 7.73% (MIR, calculated as the number of positive pools divided by the total number of mosquitoes tested and expressed as a percentage)

  1. Again, in item 3.3 of the Results section, the authors base themselves on the sample titration 3929. And the other samples that also were used in the isolation? Please, provide details of these samples.

A: Since this sample presented the highest viral load, this was the chosen sample for microscopy analysis.

  1. The samples with Ct values below 27 were selected for isolation of CxFV. However, and the other positive samples with Ct above 27? I recommend that authors indicate the average of the Cts of these samples.
  1. The Aedes spp.pools were not analyzed?

A.The average Ct value was included in item 3.2. Culex Flavivirus minimum infection rate (MIR). The Aedes spp were tested for arboviruses and CxFV, however all samples were negative. This information has been included as follows in item 3,2:

3.2. Culex Flavivirus minimum infection rate (MIR)

The pools of mosquitoes were tested for the presence of CxFV genomic material. All Aedes spp were negative.  Of the total of 134 pools from Culex spp. (401 individuals), 31 were positive with an average of Ct 26.52, giving a minimum infection rate of 7.73% (MIR, calculated as the number of positive pools divided by the total number of mosquitoes tested and expressed as a percentage)

        The authors mention that they observed a total of 31 pools of Culex spp. positive for CxFV. This is the total number of pools of Culex spp. or a part of a larger set? It is not clear in the results section, please clarify.

A: We thank you for the observation. The total number of pools was included on table 1.

  1. In the Discussion section, I would have wished to see more information on circulation of CxFV.

A: More information regarding CxFV circulation was included in this section:

“ CxFV is widely disseminated and has been found in several countries in Asia, South America, Central America, North America, and Africa. Genetically, CxFV is considered stable, and the phylogenic trees cluster geographically, even though isolation occurs from different hosts, suggesting a possible transmission between sympatric species ,[36]. The movement between continents could have happened as described by other arboviruses ,[12]. In Brazil, CxFV was reported in four geographic regions (Midwest, Northeast, Southeast, and South); however, the number and length of genomic sequences available in data banks limit robust phylogenetic analyses and dissemination studies. 

  1. I think the isolation of the virus described in the results was poorly discussed. Please improve this aspect.

A.We agree with the Reviewer, and we had complemented this item as follows:

“In this study, it was possible to isolate and titrate a sample of CxFV in C6/36 cells by PFU. As previously reported by other authors, we did not observe a CPE in the first passages in c6/36 cells  [6,8,12] . The production of characteristic CPE and formation of syncytia in C6/36 cells were described only for a few isolates of CxFV from Mexico (Blitvich, 2009)[12]. However, after the fourth passage of the isolate 3929 from Rio de Janeiro, it was possible to observe a discrete CPE and the formation of Syncytia (Figure 4); after the sixth passage, it was possible to perform the titration by PFU. Considering that the cells used for isolation came from another vector, the consecutive passages may have increased the fitness of the virus in this system.

  1. Please, include the limitations and future perspectives of your study in the Discussion section.

A.Thanks for your suggestions. The limitations were included in each topic as follows:

  1. Discussion (Third paragraph)

“However, an accurate estimation of prevalence by sex depends on more extensive sampling. Meanwhile, the large number of male mosquitoes positive for CxFV RNA contributes to the hypothesis that vertical transmission is essential for maintaining this virus in the mosquito population [11, 18, 34].”

Fourth paragraph

"In Brazil, CxFV was reported in four geographic regions (Midwest, Northeast, Southeast, and South); however, the number and length of genomic sequences available in data banks limit robust phylogenetic analyses and dissemination studies."

    A. The following excerpt regarding the perspectives was included at the end of the discussion.

"The isolation of CxFV with high viral load will allow in vivo and in vitro co-infection studies, morphogenesis analysis by electron microscopy, and evaluation of immune response by transcriptome."

Reviewer 2 Report

The work forms interesting research concerning the neglected issue, but I have concerns about this research:

1. It is very local in its impact as the samples were collected from only one very specific location.
2. Are there already cases of patients infected with arboviruses through the study area? and if the answer is yes so you showed mentioned them and if no you should discuss this issue. 

Author Response

The work forms interesting research concerning the neglected issue, but I have concerns about this research:

  1. It is very local in its impact as the samples were collected from only one very specific location.

A: We understand the reviewer's concern. However, despite this limitation, this work presents an important scientific discovery that might lead to new studies by our group and other groups. Investigations of insect-specific viruses under field conditions constitute important findings that contribute to the understanding of the evolution of Flaviviruses, and also to the emergence of viruses able to impact public health. Our study was performed in an endemic area for urban arboviruses, and the focus was on Culex sp. The Culex spp. is a vector of arboviruses such as WNV, Saint Louis Encephalitis virus, and Ilheus Virus. The presence of CxFV can modulate the circulation of arboviruses. Also, co-infection of insect-specific and arboviruses can play an important role in recombination and virus emergency events. 

  1. 2. Are there already cases of patients infected with arboviruses through the study area? and if the answer is yes so you showed mentioned them and if no you should discuss this issue. 

A: Yes, the studied area is endemic to dengue, chikungunya, and Zika viruses.

This information was included in the introduction as follows: Please find attached references of arboviruses circulation in the metropolitan region of Rio de Janeiro.

“Therefore, providing CxFV isolates becomes highly relevant to input studies of superinfection and increases understanding of vector competence's stochastic phenomenon. Since several arboviruses like Dengue, Zika and Chikungunya are endemic in Rio de Janeiro our primary goals in this study are to observe the occurrence of CxFV infection mosquitoes trapped in an urban area of Rio de Janeiro, Brazil, characterize the virus circulating, and provide isolates.”

  • de Souza TMA, de Lima RC, Solórzano VEF, Damasco PV, de Souza LJ, Sanchez-Arcila JC, Guimarães GMC, Paiva IA, da Rocha Queiroz Lima M, de Bruycker-Nogueira F, Tomé LCT, Coelho MRI, da Silva SP, de Oliveira-Pinto LM, de Azeredo EL, Dos Santos FB. Was It Chikungunya? Laboratorial and Clinical Investigations of Cases Occurred during a Triple Arboviruses' Outbreak in Rio de Janeiro, Brazil. 2022 Feb 14;11(2):245. doi: 10.3390/pathogens11020245. PMID: 35215188; PMCID: PMC8879879.

  • Xavier LL, Honório NA, Pessanha JFM, Peiter PC. Analysis of climate factors and dengue incidence in the metropolitan region of Rio de Janeiro, Brazil. PLoS One. 2021 May 20;16(5):e0251403. doi: 10.1371/journal.pone.0251403. PMID: 34014989; PMCID: PMC8136695.

Reviewer 3 Report

This manuscript describes a study whose primary goal was to observe the natural occurrence and prevalence of Culex flavivirus (CxFV) infection in Culex spp. mosquitos captured during the period from June through March in a small area of urban Rio de Janeiro, Brazil. The results document a substantial minimum infection rate (MIR) in both female and male mosquitoes, which varies over the time period examined. Examination of sequence data shows that the CxFV found belongs to Genotype II, and observations confirm previous studies showing vertical transmission of the virus and seasonality of mosquito infection. The importance of the findings will possibly be realized in future studies that determine if natural infection of a Culex mosquito with CxFV affects the outcome of subsequent infection or exposure of the same mosquito with a flavivirus capable of infecting vertebrates. It is unfortunate that these authors did not include assays for co-infection of their captured mosquitos with endemic arboviruses.

This study provides the first evidence of CxFV infection of field populations of Culex spp. mosquitoes in Rio de Janeiro. Although the study described was carried out over a small geographic area and collected a relatively small number of mosquitoes, the methods used were appropriate, well-described, and thorough.

Concerns:

  1. 2: The authors state that reference 5 presents “the first indication that it [CxFV] can produce in nature superinfection exclusion of viruses of medical interest such as West Nile [5].” Reference 5 actually challenges the hypothesis of superinfection exclusion and finds a positive association between infection of North American mosquitos with CxFV and West Nile virus (as the authors state on page 10, Discussion). Perhaps the authors should re-word the Introduction to state that reference 15 showed an initial inhibition of WNV replication in mosquitos naturally-infected with CxFV.

Minor:

The authors should strive for consistency in italicization, capitalization, and use of a period/point vs. comma in numbers. Examples:

  1. 2 and 10: Culex pipiens and Cx. quinquefasciatus should be italicized.
  2. 2: Culex Pipiens species name should not be capitalized.
  3. 6 and 7: 3,875 should be 3.875 and 7,730673 should be 7.730673

Author Response

  1. 2: The authors state that reference 5 presents “the first indication that it [CxFV] can produce in nature superinfection exclusion of viruses of medical interest such as West Nile [5].” Reference 5 actually challenges the hypothesis of superinfection exclusion and finds a positive association between infection of North American mosquitos with CxFV and West Nile virus (as the authors state on page 10, Discussion). Perhaps the authors should re-word the Introduction to state that reference 15 showed an initial inhibition of WNV replication in mosquitos naturally-infected with CxFV.

A: We are grateful for your important note. This excerpt of the introduction was revised as follows, and the reference was cited:

Original: “Culex Flavivirus (CxFV) is a classical insect-specific virus, which arouses interest after the first indication that it can produce in nature superinfection exclusion of viruses of medical interest such as West Nile [5].

Correction: “Culex Flavivirus (CxFV) is a classical insect-specific virus, which arouses interest after the first indication that colonies of Culex Pipens mosquitoes naturally infected by CxFV presented a delayed dissemination of West Nile virus on vector competence study” [5]

[5] Bolling,B.G.;Olea-Popelka,F.J;Eisen, L.;Moore,C.G.;Blair,C.D. Transmission dynamics of an insect-specific  flavivirus  in  a  naturally  infected  Culex  pipiens  laboratory  colony  and  effects  of  co-infection  on vector competence for West Nile virus. Virology. 2012,427(2):90-7. https://doi.org/ 10.1016/j.virol.2012.02.016

 Minor:

 The authors should strive for consistency in italicization, capitalization, and use of a period/point vs. comma in numbers. Examples:

  1. 2 and 10: Culex pipiens and Cx. quinquefasciatus should be italicized.
  2. 2: Culex Pipiens species name should not be capitalized.
  3. 6 and 7: 3,875 should be 3.875 and 7,730673 should be 7.730673

A: We thank you for this observation. The manuscript was fully revised for consistency in italicization, capitalization, and use of a period/point vs. comma in numbers.
